# Effects of Open and Forest Habitats on Distribution and Diversity of Bumblebees (*Bombus*) in the Małopolska Upland (Southern Poland): Case Study

**DOI:** 10.3390/biology10121266

**Published:** 2021-12-03

**Authors:** Jolanta Bąk-Badowska, Anna Wojciechowska, Joanna Czerwik-Marcinkowska

**Affiliations:** 1Institute of Biology, Jan Kochanowski University, 25-406 Kielce, Poland; jolanta.bak-badowska@ujk.edu.pl; 2Faculty of Biology and Environmental Protection, Nicolaus Copernicus University, 87-100 Toruń, Poland; ankawoj@umk.pl

**Keywords:** *Bombus*, community ecology, conservation, insect pollinators, landscape ecology, Europe

## Abstract

**Simple Summary:**

Southern Poland represents one of the most diverse habitats for bumblebees (*Bombus* sp.); however, little is known about the abundance and distribution of many insect species in this region. Bumblebees are important for crop and wildflower pollination in different temperate latitudes because many plant species are only pollinated by them. Studies were conducted in natural and semi-natural habitats in southern Poland during the years 2003–2006 and compared with material collected from 2017–2020. During this eight-year-long study, more than 6214 bumblebee specimens of 25 species were found in the Małopolska Upland. The most frequently observed bumblebee species were: *Bombus pascuorum, B. lapidarius, B. pratorum,* and *B. lucorum.* The low-numbered bumblebees were: *Bombus humilis, B. pomorum, B. veteranus, B. muscorum,* and *B. semenoviellus*. There were also four rarely found species: *Bombus confuses, B. ruderatus, B. soroeensis,* and *B. jonellus*.

**Abstract:**

Bumblebees are an important insect group occurring in different land ecosystems, but the number of these species has declined dramatically across Poland as well as in Europe in recent years. The fragmentation of bumblebee habitats influences the abundance and richness in community composition and trophic and competitive interactions. During the years 2003–2006 and 2017–2020, we studied the diversity and distribution of bumblebee species in two natural (boron-mixed *Vaccinio-Piceetea* and riparian forest *Querco-Fagetea*) and two semi-natural (segetal-ruderal *Stellarietea mediae* ruderal *Artemisietea vulgaris*) habitats in southern Poland. For that, we evaluated how habitats as well as local flowering communities influenced bumblebees’ abundance, richness, and community composition in 16 sites (which are located in four parks). Bumblebee communities responded to environmental factors in different ways according to the type of habitat. Vegetation factors were the most important drivers of bumblebee community structures. Forests showed the lowest bumblebee abundance, richness, and diversity, and the highest dominance levels of these parameters were found in the open ruderal-segetal habitats. The meadows from the *Molinio arrhenatheretea* class were characterized by bumblebee communities with a more complex structure. Species diversity was positively correlated with open ruderal-segetal habitats, and negatively with mixed forest cover, while abundance was positively correlated with forest cover. Studies like this are necessary to anticipate the impact of habitat fragmentation on bumblebee decline.

## 1. Introduction

Bumblebees play an important role as pollinators of many crop plants and wildflowers. Although the distribution of bumblebees encompasses a wide geographic range from Arctic tundra to lowland tropical forest, they are clearly most abundant in mountain habitats and the cold and temperate regions of the Northern Hemisphere [1,2]. Currently, more than 250 species are known within the genus *Bombus* to occur on all continents except Antarctica [3]. In Central Europe, 40 species have been described, while in Poland, there are 37 bumblebee species, including 10 species of cuckoo bees (*Psithyrus*), and 4 species are sporadically encountered or regionally extinct [4,5,6]. Many bumblebee species are still poorly known [7,8,9], and the causes of extinction of bumblebees have been debated for over 60 years [3,9,10]. All bumblebee species are protected throughout Poland, but their numbers are decreasing every year [4].

As among many wild bees, bumblebee abundance and diversity declined in recent years, which may threaten the stability of pollinator communities [11,12,13]. Ollerton et al. [2] observed that 90% of flowering plant species require animal pollination. Bumblebees and bees are bonded to flowers by their use of pollen as a protein source, hence they are the most important pollinator species in terrestrial ecosystems [13,14]. The numbers of bumblebee species are declining in Europe, North America, and Asia due to a number of factors, including land-use change (reduced food plants), loss of nesting habitats, and climate change effects [15,16,17]. This effect was also observed in boron-mixed (*Vaccinio-Piceetea*) and riparian forest (*Querco-Fagetea*), which are some of the most valuable and endangered ecosystems in Central Europe, and their conservation is of worldwide importance for preserving plant and animal biodiversity [18].

Although knowledge of bumblebees has been increasing over the last few decades, habitats in some geographic areas remain poorly studied. Poland is one of the European countries which lacks comprehensive data on distribution and diversity of bumblebee species. However, the bumblebees of the Małopolska Upland were already studied in the 1960s and 1980s [19,20]. The scientific monitoring of our current study was aimed at evaluating the distribution and diversity of bumblebees in the Małopolska Upland (southern Poland) collected during the years 2003–2006 and 2017–2020. Another goal of our study was to determine the environmental factors, as well as their influence on bumblebee diversity in natural and semi-natural habitats.

## 2. Materials and Methods

### 2.1. Study Site Description

During the years 2017–2020, we conducted a survey of the bumblebee species at 16 sites in the Małopolska Upland (southern Poland) previously sampled between 2003–2006. In each of the 16 sites, we established a permanent bee walk transect of 100 m to 100 m (squares of the UTM grid) to count foraging bumblebees (a modified version of Goulson et al. [6]). The field trips covered the entire flowering season (from the beginning of May to the beginning of September), and we collected bumblebees with an aerial net on flowers or while in flight. GPS locations were recorded using handheld GPS units or cell phones (Garmin Ltd., Apple Inc., Modesto, CA, USA) and later verified using Google Earth Pro (version 7.3.2. Google LLC, CA, Mountain View, USA). The collecting sites are shown on the map of the Małopolska Upland (Figure 1). These sites were located in four parks: Świętokrzyski National Park (SNP), Cisowsko-Orłowiński Landscape Park (COLP), Chęcińsko-Kielecki Landscape Park (ChKLP), and Nadnidziański Landscape Park (NLP). Based on phytosociological and ecological analysis of forest and non-forest communities occurring in southern Poland, we distinguished: (a) association *Querco roboris-Pinetum*, class *Vaccinio-Piceetea,* with dominant species *Pinus sylvestris* (pine)*, Quercus robur* (common oak)*, Betula pendula* (silver birch)*, Frangula alnus* (alder buckthorn) and *Corylus avellana* (common hazel); (b) association *Fraxino-Alnetum,* class *Querco-Fagetea* with dominant species *Alnus glutinosa* (common alder), *Filipendula ulmaria* (meadowsweet), *Equisetum sylvaticum* (wood horsetail)*, *Asarum europaeum** (asarabacca), *Lysimachia vulgaris* (yellow loosestrife), and *Valeriana officinalis* (garden valerian);
(c) class *Stellarietea mediae* with segetal and meadow species and dominant field and ruderal weeds, mainly grasses; and (d) class *Artemisietea vulgaris* with dominant ruderal species *Artemisia vulgaris* (common mugwort), *Agrostis capillaris* (common bent)*, Cirsium arvense* (creeping thistle), and *Plantago lanceolata* (ribwort plantain) and *Vicia hirsuta* (hairy tare) (Figure 2).

### 2.2. Collections

The bumblebee species were harvested every two weeks from mid-May to mid-September, between 9.00 a.m. and 4.00 p.m. on days without precipitation, with little or no wind, and with air temperature above 18 °C (Appendix A). During each visit, all foraging bumblebees within a transect, as well as the plants on which they were observed, were registered. We identified plants using the botanical keys [21,22,23]. Single specimens of bumblebee were caught with an entomological net and photographed live, free-foraging. The specimens were prepared by using cyanide or ethyl acetate. All bumblebees were placed in an airtight container with a few layers of tissue and the addition of a few drops of ethyl acetate. After the field sampling, the specimens were dry-mounted on standard insect pins for identification. In total, 1536 samples were collected from 2017–2020 and deposited in the Institute of Biology (Jan Kochanowski University, Kielce) entomological collections following analysis. The community structure, species relationships, foraging activities, abundance, and phenology of every species were studied throughout the season. Species identifications made in the field were verified in the laboratory using the taxonomic keys of Krzysztofiak et al. [24], Pawlikowski [25], Dylewska and Flaga [26], Dylewska et al. [27], Williams et al. [28], and Banaszak [29].

In each habitat, soil physical (density, structure, and texture) and chemical (inorganic and organic matters) properties were studied using a multifunctional probe YSI Professional Plus. For a detailed analysis of the chemical parameters, water samples (500 mL) were taken, which were conserved with chloroform (CHCl_3_) and stored at −10 °C for further analysis.

### 2.3. Statistical Analysis

The Simpson diversity index was calculated based on the data on the occurrence of insects in the studied sites. One-way ANOVA was used to test for changes in Simpson diversity at sites across years. The calculations of the index were made in the PAST program [30]. One-way ANOVA and graphs considering the mean value of the indicator, standard error, and standard deviation were calculated using the Statistica 9.0 software (TIBICO Software Inc., Palo Alto, CA, USA) [31]. The data on the number and occurrence of insects were also used to perform a direct ordination analysis (RDA). The file with environmental data was constructed on the basis of information about the year of research, the community, and the facility where the research was conducted. This information was coded in the “0–1” system. In order to determine which of the resulting variables were statistically significant for the diversity in insect occurrence, the forward-selection and a Monte Carlo permutation test were performed during the RDA. The result of the RDA is an ordinance diagram in which both species and environmental variables are marked with vectors. Changes in the number of species in samples in the ordination space were presented using a diagram. In order to indicate the tendency of species occurrence in sites and time, indirect coding analyses (Principal Component Analysis, PCA) were also performed. Ordinance analyses were done using the Canoco 5.0 software (Ithaca, NY, USA) [32].

## 3. Results

Twenty–five species of bumblebees were collected in the Małopolska Upland during regular field trips from 2003–2006 and from 2017–2020. A total of 3671 bumblebee specimens, between 2003 and 2006, and 2543 bumblebee specimens from 2017–2020 in the natural and semi-natural habitats were identified, respectively (Table 1). The most frequently observed bumblebee species (recorded in more than 50% of the examined UTM transects) were: *Bombus pascuorum, B. lapidarius, B. pratorum* and *B. lucorum.* The group of medium-numbered bumblebees (30–50% of UTM transects) includes four species: *Bombus ruderiarius, B. sylvarum, B. hortorum,* and *B. hypnorum.* The low-numbered bumblebees were: *Bombus humilis, B. pomorum, B. veteranus, B. muscorum,* and *B. semenoviellus*. There were also four rarely found species (in less than 10% of UTM transects): *Bombus confusus, B. ruderatus, B. soroeensis,* and *B. jonellus*. A maximum of 25 species were registered in the transect. Eleven species of bumblebees were collected in Świętokrzyski National Park in 2004, while 15 species were observed in 2006. In the Cisowsko-Orłowiński Landscape Park, we found 21 species of bumblebees in 2020, but 15 species in 2003, and 19 in 2004. Whereas in the Chęcińsko-Kielecki Landscape Park, we observed 12 bumblebee species in 2004 to 17 species in 2006, and in Nadnidziański Landscape Park were from 14 in 2004 to 18 in 2006, but in 2020 24 species were found. The bumblebee species prefer plants belonging to the Lamiacae and Fabaceae families. However, *B. terrestris, B. lapidaries,* and *B. pratorum* were more commonly found on the Asteraceae familly. Sixty-five plants were the food base for the bumblebee, such as *Echium vulgare, Lotus corniculatus, Trifolium arvense, T. pratense,* and *Viccia cracca*. A total of 1083 blooming plants were identified. The species most frequently collected were *Picea* sp. (46.6% of specimens were collected on plants), *Taraxacum officinale* (58%), and *Salix* sp. (42%). The studies show that the number of bumblebees species changed over the years and, based on phytosociological and ecological analysis of plant communities, the flower richness also changed (for plant species we estimated flower richness as the average number of flowering species per sampling day and site). We registered a total of 1370 bumblebee specimens in Cisowsko-Orłowiński Landscape Park and 498 specimens in Świętokrzyski National Park between 2003–2006. Bumblebee abundance decreased between 2017 and 2020. Overall, we recorded 1984 bumblebee specimens in Cisowsko-Orłowiński Landscape Park, and only 328 specimens in Świętokrzyski National Park.

For each of the study samples, the Simpson diversity index varied from 0 (one species in one of the samples) to 0.895 (Figure 3) for either the year, community, or study site. In no case were the differences in the index values statistically significant. The analysis of redundancy indicated that the study years (2005, 2006, and 2020), the communities (semi-natural A1 and A2) and the study site (ChKLP, ŚNP, and COLP) were important for the differentiation in the number of bumblebees (Figure 4). These variables account for 40.8% of the total variability of the study set. The significance of the variables also results from the relationship of some species with the studied variables; for example, *Bombus pascuorum* was the most numerous in 2006. Some species were present only in specific sites; for example, *B. pratorum* and *B. sylvestris* were most numerous in the COLP. The number of species collected from these two parks was also significantly higher in 2005 and 2006, compared with the other sites (ANOVA, *p* < 0.05). More species also occurred in semi-natural habitats such as on the fallow. This regularity became apparent when isolines indicating the number of species were placed in the ordination space (Figure 5). The variables attached with a large number of species in the samples and in ŚNP, where the species are significantly lower, are statistically significant.

The analysis of the indirect ordinance was presented three times to indicate regularities in the occurrence of bumblebee species depending on the years (Figure 6), sites (Figure 7), and habitats (natural and semi-natural) where the study was performed (Figure 8). The *B. pascuorum* and *B. hortorum* were recorded more frequently in 2005 and 2006 than later years. It was observed that especially in 2004, the occurrence of these species was sparse (Figure 6). Species from the *Pseudobombus* family appeared sporadically in the study years, while in 2020, *B. campestris* and *B. sylvestris* were more numerous in semi-natural habitats.

The species *B. subterraneus* and *B. humilis* were most often found in NLP and ChKLP parks, wheras *B. biohemicus, B. norvegicus, B. pratorum,* and *B. sylvestris* dominated in COLP (Figure 7). The occurrence of *Pseudobombus* species e.g., *B. bohemicus, B. campestris, B. sylvestris,* and *B. norvegicus* were limited to COLP. Some species, such as *B. barbutellus* and *B. vestalis* were also recorded in SNP. Species such as *B. biohemicus, B. norvegicus, B. pratorum,* and *B. sylvestris* preferred the forest (N1, N2) and open (A1, A2) habitats (Figure 8). The *Pseudobombus* species are dominant also in natural and semi-natural habitats. Representatives of these groups such as: *B. bohemicus, B. campestris, B. sylvestris, B. norvegicus* prefer natural habitats. In the semi-natural habitats, a few species with low numbers (e.g., *B. barbutellus, B. vestalis*) were found (Figure 9). The *Bombus* family prefers open, ruderal, and semi-natural habitats. In forest communities (N1, N2) numerous occurrences of *B. pratorum* species were recorded (Figure 8).

Two hundred and fifty-five specimens of bumblebee were found in natural habitats in 2003, while in 2004 there were 259 specimens, and 521 specimens of bumblebee in 2005, whereas in 2006, there were 698 specimens (Table 2). Meanwhile, in the year 2017, 369 specimens of bumblebee were identified, and in the following years of the study, a decline in the bumblebee communities was observed.

## 4. Discussion

In this study, bumblebee diversity and distribution in the Małopolska Upland (southern Poland) in natural and semi-natural habitats during 2003–2006 and 2017–2020 were presented. The relationships between bumblebee community and habitat, study years, and species occurrence in parks were observed. During this eight-year study, we observed that the number of flowering plants and bumblebee abundance and richness decreased the composition of the communities changed, by reducing habitat-specialized species in favor of highly generalist ones. The greatest bumblebee species diversity was found in NLP (24 species), while in ChKLP, there were 23 species. Species such as: *B. pascuorum, B. terestris,* and *B. lucorum* occurred in natural and semi-natural habitats, in all studied years. *B. hypnorum* is considered a species typical from northern forests and, normally, nests above ground, mainly in tree cavities and in the wooden nesting boxes for birds. For bumblebee species, in study habitats, the number of flowering plants was important, as it is positively related to their abundance and richness. Bumblebees preferred plants such as: *Caluna vulgaris*, *Centaurea jacea*, *Centaurea scabiosa*, *Lamium album*, *Rubus hirtus*, *Trifolium pratense* and *Trifolium alba,* and *Vicia cracca*. Moreover, Westphal et al. [33] and Person et al. [34] described the relationship of bumblebee abundance and richness with flower density varied along the flowering season. The smallest quantity of species was captured in the COLP (21) and in the SNP (only 17). Natural and semi-natural habitats of SNP are poor in flowers and flower availability plays such an important role in bumblebee distribution. The number of described species from the Małopolska Upland also correlates with the research of Krzysztofiak [35], which studied 23 species of bumblebees in Wigierski National Park, whereas Pawlikowski [36] described only 21 species in pine stands in the Toruń Basin. It was observed that along with abundance and richness, the composition of the bumblebee communities also changed during the eight-year study. We concluded with Gomez-Martinez [37] that species adapted to forest habitats decreased in number with forest fragmentation, while species related to open areas become more abundant.

Nieto et al. [38] stated that, among European bees, the genus *Bombus* includes the highest percentage of species with an extinction risk according to IUCN criteria. Among the 68 species in Europe, 45.6% are stable, and 13.2% expose positive population trends and an expansion of their distribution [38]. The community of bumblebees occurring in the Małopolska Upland (southern Poland) needs permanent scientific monitoring. According to IUCN criteria [38,39], among the 30 bumblebee species found in Poland, 19 species were passed in the Polish Red List [40], 11 species labelled as vulnerable (VU), 6 as poorly recognized status (DD), and 2 as endangered (EN).

Sarospataki et al. [41] concluded that except for the UK, in most of Europe, detailed information on the abundance and distribution of bumblebee species is not available. The current status of pollinators in Central Europe, such as Carpathian Basin, is not well-known [38]; however, the fauna and ecology of bumblebees in the European North are quite well-studied [42]. More than half of the Małopolska Upland bumblebee species were found to be rare or moderately rare. The very similar situation is in Hungary [41], Belgium [42] and Bawaria [6]. Four bumblebee species from the Małopolska Upland were rare.

*Bombus confusus* Schenck is protected in Poland [40]. The distribution of this species is Europe and Central Asia [43]. This species does not reach the Mediterranean peninsulae, nor the Mediterranean sea coast. It is assessed as vulnerable in the IUCN Red List of European Bees [38]. In Poland, *B. confusus* is rare, mainly both on the edges of forests, warm meadows, and fields. It nests mainly in the ground. The species is endangered by mechanized forest management, the use of chemicals to combat pests of crops, burning and ploughing of balks, roadsides, removal of brushwood, fallow land, and increased tourism in Poland. The only observation of this species is from Nadnidziański Landscape Park.

*Bombus ruderatus* (Scopoli) is a large garden or ruderal bumblebee species, common in western Europe, especially in its Mediterranean zone. This species is very rare in Poland [40] and it was reported from three sites in Świętokrzyski National Park and Chęcińsko-Kielecki Landscape Park. The eastern limit of its range is less clear, though it includes Poland, after Hungary and Slovakia, and seems to reach Ukraine [44]. *Bombus soroeensis* (Fabricius) Palearctic is a species widespread in Europe (from Spain to the southern part of Central Siberian Plateau and Turkey, and the Carpathian), but highly localized [45]. This species is vulnerable in Poland [40] and known from single sites. It was found on the smaller-flowered legumes, such as *Melilotus* sp., and *Campanula* sp. It was collected from two sites in Świętokrzyski National Park and Nadnidziański Landscape Park. During our studies, *B. soroeensis* was found on xerothermic grasslands and in forest environments.

*Bombus jonellus* (Kirby) is a small species, widespread and common in Europe: from Iceland and the Sierra Cantabrica in the west, to the Anadyr on the Pacific. In the south of Europe, the range of this species is restricted to montane biotopes, reaching the highest mountains of the Iberian Peninsula where it is very rare [38]. This species occurs on the moorland (on the Ericaceae) and it is vulnerable in Poland [40]. During our studies, we observed that the most frequently visited plant species by *B. jonellus* were wild thyme (*Thymus praecox*), marsh cinquefoil (*Comarum palustre*), water avens (*Geum rivale*), and tufted vetch (*Vicia cracca*). This species was collected from two sites in Nadnidziański Landscape Park and Cisowsko-Orłowiński Landscape Park.

Rasmont and Mersch [13] and Rollin et al. [42] suggested that among European bumblebees, the severely declining species tend to be those with a low genetic diversity, a short flight season, a late emergence, a small number of habitat types, a long tongue, and a restricted dietary breadth associated a narrow pollen diet or with flowers with long corolla such as *Fabaceae,* such as *B. humilis, B. ruderatus,* and *B. subterraneus.* However, other studies [4,6,39,42] indicated that the climate may have an influence on complex interactions between ecological traits and environmental factors that may be associated with higher susceptibility of bumblebee declines. A similar situation has occurred with Hungarian bumblebees, where seven species are critically endangered (CR), three are endangered (EN), and two species are vulnerable (VU). *B. soroeensis* as a vulnerable species and *B. subterraneus* shows a frequency trend, but both are rare, and present in the IUCN Red List of European Bees [38].

Rollin et al. [42] concluded that traits correlated with higher rates of species extinction are a narrow geographic distribution, slower reproductive rate, low population density, and ecological specialization. Protective measures concerning bumblebees should mainly be the protection of the natural habitats and natural resources, which are the optimal place for the development of many species. The preservation of bumblebees is possible due to supplementing the food base by sowing attractive plants and planting pollen-bearing shrubs and trees, protection of forest islands, woodlands, and roadsides.

Several studies [46,47] documented that many environmental factors, such as temperature, wind, sun exposure, and humidity, affect the activity of bumblebees and their occurrence in different habitats. Our results indicate that the diversity of flowering plants occurring in natural and semi-natural studied habitats, as well as local features of the site itself are important for bumblebee diversity and abundance. Westphal et al. [33] implicated the importance of the habitats in maintaining bumblebee abundance and finding a correlation between the availability of flowering crops and bumblebee density. Rasmont et al. [27] suggested that climate change poses a threat to many bumblebee species worldwide and to change the structure of their functioning, including access to host plants. On the other hand, the expansion of some bumblebee species into new areas has been observed [48,49]. Distribution and occurrence data of bumblebees in new habitats indicates the ecological plasticity of these species and the possibilities of adaptation in the context of ongoing changes.

This study conducted in Poland indicates new locations of bumblebees and their dispersal, especially in the southern part of the country. Michołap et al. [50] suggested that the increase of continentalism in Europe could also be likely to affect the current status of bumblebee species expansiveness, while the biodiversity of plants occurring in Central and Eastern Europe may help these species spread throughout Europe.

In our research, we noticed that forest specialist bumblebee species exist, where Scoble [51] also found forest bees and forest butterflies which are considered creatures of open habitats [52], although in temperate zones, butterflies tend to be associated with open areas. We have suggested that the type of habitats might be an important determinant of pollinator responses to land use change with, for example, forest to open transitions having different effects from open-to-open transitions, as described by Winfree et al. [53]. Comparing habitat types, pollinator abundance and species richness in natural habitats, such as the forest, were often lower than in anthropogenic habitats. Steffan-Dewenter et al. [54], Sjodin et al. [55], and Winfree et al. [53] concluded that many studies use semi-natural habitats, such as grazed grasslands, fallow agriculture, and suburban gardens as the good bumblebees’ occurrence habitat. These semi-natural habitats are not then compared to natural habitats, such as the forest. However, the loss of semi-natural habitats has negative effects on pollinators of various species [56]. Tews et al. [57] concluded that low-level semi-natural land use may increase the heterogeneity of habitats and resources, thus increasing niche diversity. Klein et al. [58] observed that pollinators could nest in forest habitats, but forage in semi-natural habitats. Zajdel et al. [59] concluded that the size of parks, percentage of area covered by trees, and characteristics of the areas surrounding the parks were not significant for the diversity and abundance of bumblebees, however indicating the importance of the semi-natural habitats for the species diversity. Meanwhile, Diaz-Forero et al. [60] found that the presence of forest is very important for bumblebees, even for those species that seem to prefer open areas, because forest habitats may provide overwintering sites and nesting places.

Despite the overall decline in bumblebees, not all the species may respond similarly to landscape fragmentation. Studies have shown that while some species have considerably declined in fragmented habitats, others have remained relatively abundant. Habitat fragmentation can affect the diversity of bumblebee species and communities, and may depend, among other aspects, on their habitat preferences, foraging ranges, and behavioural patterns [37].

Bumblebees do not show so-called floral fidelity to all flowers encountered on their flight path and do not focus, like bees, on just one species that blooms en masse at a time [61]. Sikora et al. [18] suggested that bumblebee species can also be treated as bioindicators of the state of the natural environment, and natural and anthropogenic habitats (e.g., urban space) that can be a refuge for bumblebees and many other species of insects involved in the process of pollination of plants. In general, bumblebee species diversity, and especially common species, can be promoted in different kinds of landscapes by ensuring a variety of good-quality local habitats [62]. Rollin et al. [42] concluded that long-term records are necessary to estimate population trends accurately and to propose appropriate mitigation strategies. The Polish *Bombus* fauna needs much more protection and higher conservation efforts than they are benefiting from today.

## 5. Conclusions

The paper presented the results of scientific monitoring of bumblebees carried out within the Małopolska Upland in the years 2003–2006 and 2017–2020. Twenty–five species of bumblebees were found on 16 sites. Based on an original data set of 6214 specimens, we also assessed a high proportion of species declining, analyzing both richness changes and species range sizes during the last 8 years. Our results indicated that the natural and semi-natural resources habitats in park areas are important for bumblebee diversity and abundance in the Małopolska Upland (southern Poland). Providing flowering areas might enhance the diversity and abundance of bumblebees, as well as other insect pollinators. Plant species diversity and composition are the most important factors determining bumblebee abundance and diversity.

## Figures and Tables

**Figure 1 biology-10-01266-f001:**
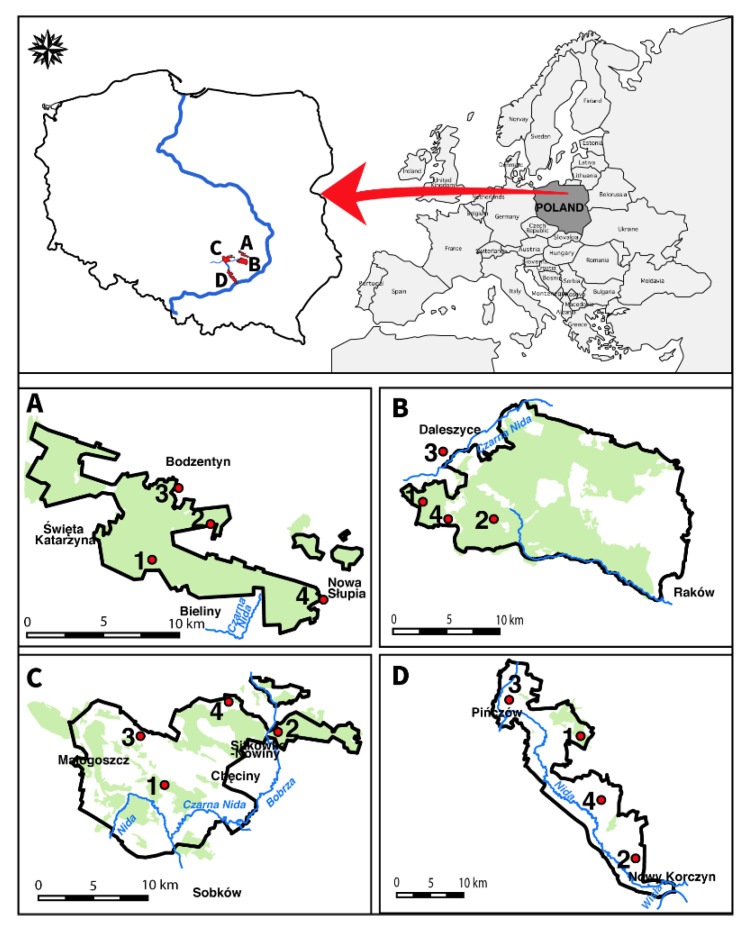
Bumblebee sampling sites in the Małopolska Upland: Map of Poland showing the geographic location of four parks where bumblebee specimen collections took place (**A**) Świętokrzyski National Park; 1. *Querco roboris-Pinetum* with class *Vaccinio-Piceetea*, UTM: DB 93, 2. *Fraxino-Alnetum* with class *Querco-Fagetea,* UTM: DB 99, 3. *Stellarietea mediae*, UTM: DB 94, 4. *Artemisietea vulgaris,* UTM: EB 03; (**B**) Cisowsko-Orłowiński Landscape Park; 1. *Querco roboris-Pinetum* with class *Vaccinio-Piceetea*, UTM: DB 82, 2. *Fraxino-Alnetum* with class *Querco-Fagetea*, UTM: DB 82, 3. *Stellarietea mediae*, UTM: DB 82, 4. *Artemisietea vulgaris,* UTM: DB 82; (**C**) Chęcińsko-Kielecki Landscape Park; 1. *Querco roboris-Pinetum* with class *Vaccinio-Piceetea*, UTM: DB 52, 2. *Fraxino-Alnetum* with class *Querco-Fagetea*, UTM: DB 63, 3. *Stellarietea mediae*, UTM: DB 53, 4. *Artemisietea vulgaris*, UTM: DB 63; (**D**) Nadnidziański Landscape Park; 1. *Querco roboris-Pinetum* with class *Vaccinio-Piceetea*, UTM: DB 79, 2. *Fraxino-Alnetum* with class *Querco-Fagetum*, UTM: DB 68, 3. *Stellarietea mediae,* UTM: DB 69, 4. *Artemisitea vulgaris,* UTM: DB 78. The locations are projected from GPS data to a SRTM elevation data set.

**Figure 2 biology-10-01266-f002:**
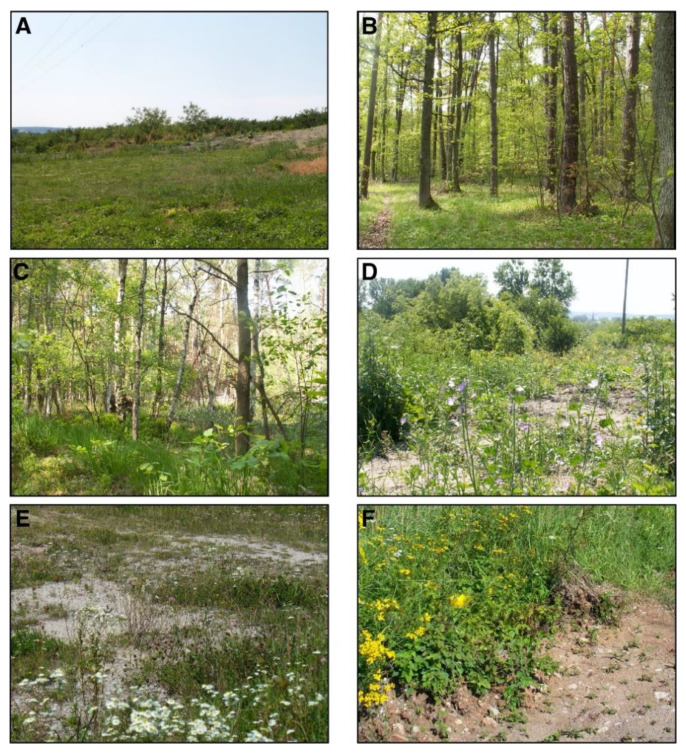
The representative natural and semi-natural habitats in the Małopolska Upland (southern Poland) where bumblebee collections were carried out between 2003–2006 and 2017–2020: (**A**) ruderal plant communities from the class *Stellarietea mediae* (NLP); (**B**) boron-mixed *Vaccinio-Piceetea* (ŚNP); (**C**) riparian forest *Querco*-*Fagetea* (ŚNP); (**D**–**F**) class *Artemisietea vulgaris* with all ruderal and nitrophilous fringe communities dominated by biennials and perennials (COLP).

**Figure 3 biology-10-01266-f003:**
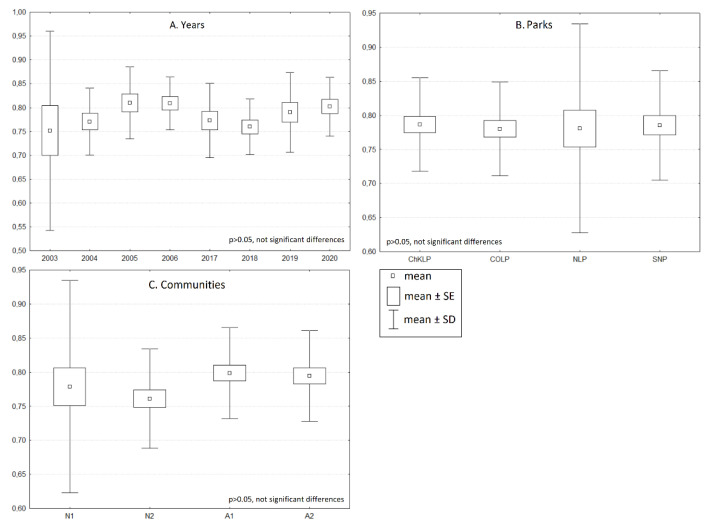
Simpson’s diversity index calculated for: (**A**) the years in which the studies was performed; (**B**) the research objects–parks; (**C**) the studied communities. The differences were not statistically significant in any case. Legend: N1, N2-natural habitats; A1, A2-semi-natural habitats. Sites: ChKLP–Chęcińsko-Kielecki Landscape Park; COLP–Cisowsko-Orłowiński Landscape Park; NLP–Nadnidziański Landscape Park; SNP–Świętokrzyski National Park.

**Figure 4 biology-10-01266-f004:**
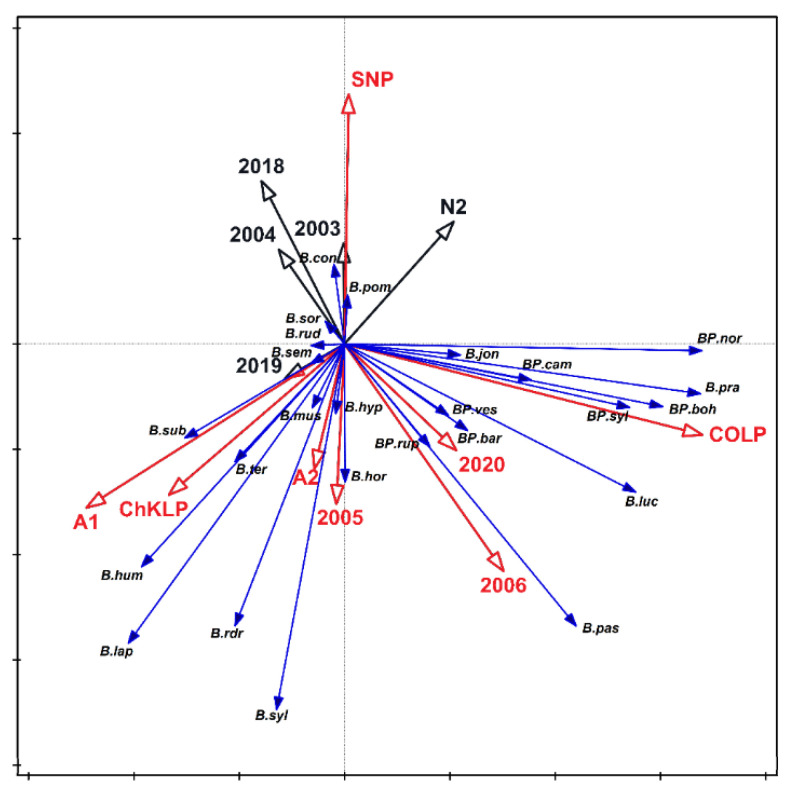
Habitat fragmentation and bumblebee community composition. Redundancy analysis showing the relationships between bumblebees (species), habitats (natural N1, N2 and semi-natural A1, A2), sites (ChKLP—Chęcińsko-Kielecki Landscape Park, COLP—Cisowsko-Orłowiński Landscape Park, NLP—Nadnidziański Landscape Park, SNP—Świętokrzyski National Park), and years of the study. The vectors of variables significantly differentiating the examined sets are marked in red. The variables account for 40.8% of the total variability. Species were abbreviated and labelled with blue, smaller, vectors. Abbreviations of species names in Figure 4 consist of one letter of a generic name and three letters of a species name. Legend: *BP.bar Bombus (Ps.) barbutellus; BP.boh B. (Ps.) bohemicus; BP.cam B. (Ps.) campestris; BP.nor B. (Ps.) norvegicus; BP.rup B. (Ps.) rupestris; BP.syl B. (Ps.) sylvestris; BP.ves B. (Ps.) vestalis; B.con B. confusus; B.hor B. hortorum; B.hum B. humilis; B.hyp B. hypnorum; B.jon B. jonellus; B.lap B. lapidarius; B.luc B. lucorum; B.mus B. muscorum; B.pas B. pascuorum; B.pom B. pomorum; B.pra B. pratorum; B.rdr B. ruderarius; B.rud B. ruderatus; B.sem B. semenoviellus; B.sor B. soroeensis; B.sub B. subterraneus; B.slv B. sylvarum; B.ter B. terrestris*.

**Figure 5 biology-10-01266-f005:**
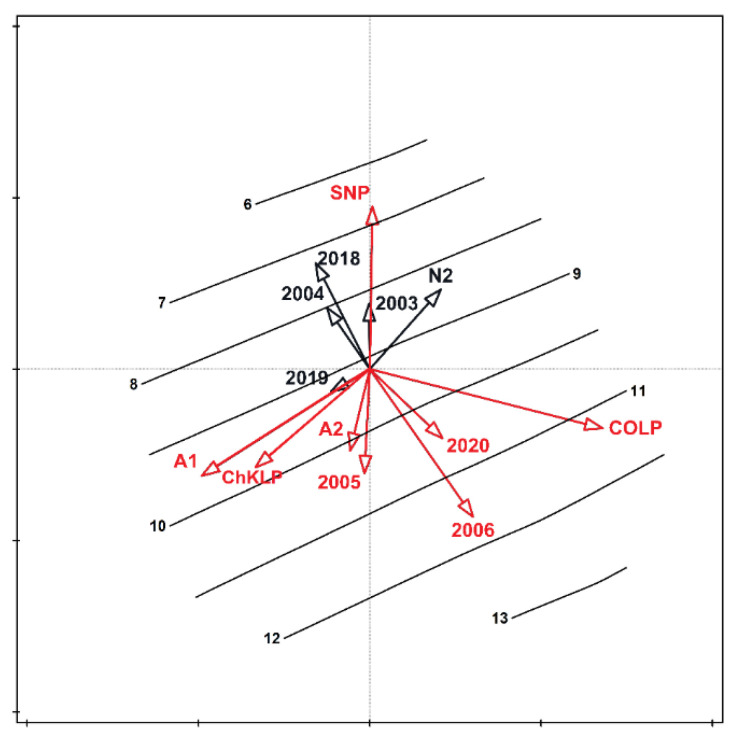
Average number of bumblebees in the ordination space. The vectors of variables significantly differentiating the examined set are marked in red. Isolines represent the gradient of the changing number of species in the tested samples. Legend: N1, N2—natural habitats; A1, A2—semi-natural habitats. Sites: ChKLP—Chęcińsko-Kielecki Landscape Park; COLP—Cisowsko-Orłowiński Landscape Park; NLP—Nadnidziański Landscape Park; SNP—Świętokrzyski National Park.

**Figure 6 biology-10-01266-f006:**
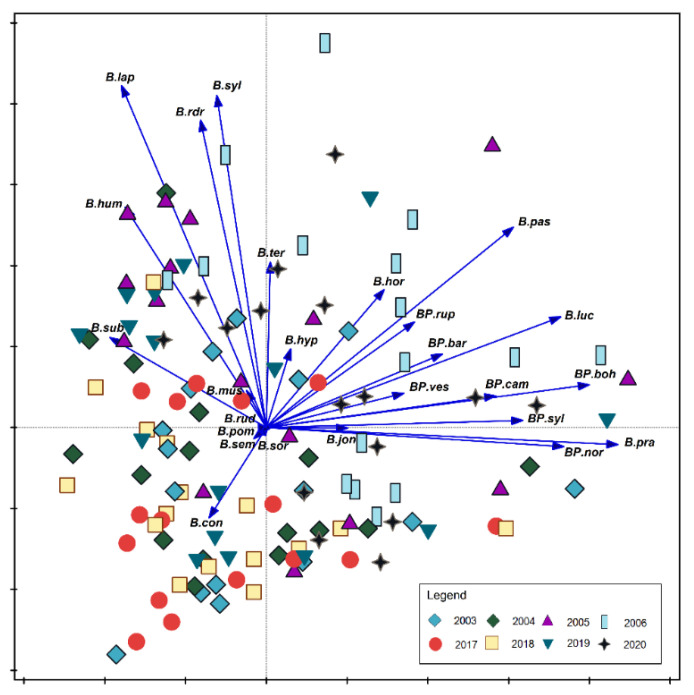
Principal Component Analysis (PCA) showing the relationships between bumblebees community and study years. The species are presented with vectors, and study years are marked with geometrical figures according to the legend in the Figure: blue rhombus—2003, green rhombus—2004, purple triangle—2005, light blue rectangle—2006, red circle—2017, green triangle, brown cross—2020. Abbreviations of species names consist of one letter of a generic name and three letters of a species name. All abbreviations as in Figure 4.

**Figure 7 biology-10-01266-f007:**
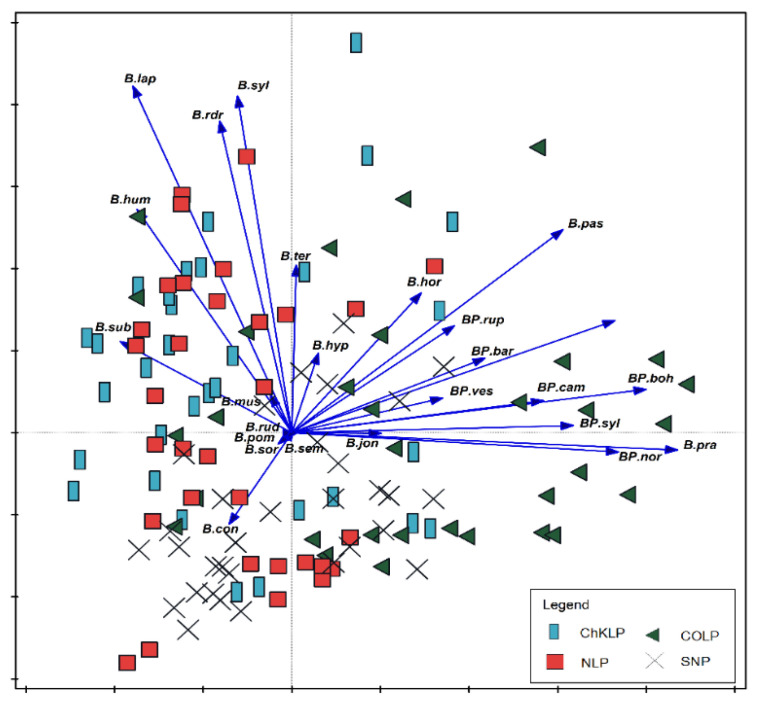
Principal Component Analysis (PCA) showing the relationships between bumblebees community and occurrence in parks. The species are presented with vectors and the parks are marked with geometrical figures: rectangle—ChKLP (Chęcińsko-Kielecki Landscape Park), triangle—COLP (Cisowsko-Orłowiński Landscape Park), square—NLP (Nadnidziański Landscape Park), cross—SNP (Świętokrzyski National Park). Abbreviations of species names consist of one letter of a generic name and three letters of a species name. All abbreviations as in Figure 4.

**Figure 8 biology-10-01266-f008:**
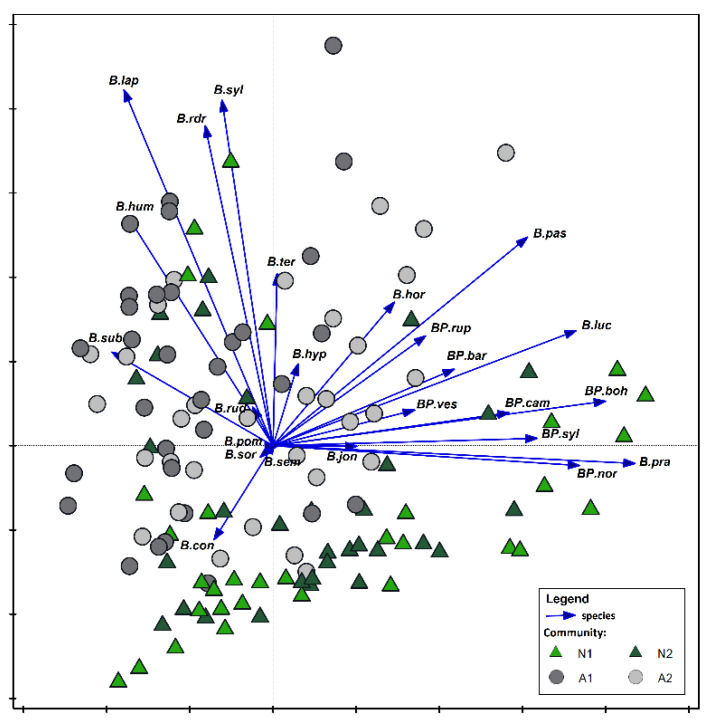
Principal Component Analysis (PCA) showing the relationships between bumblebees community and natural (A1) and semi-natural (A2) habitats. The species are presented with vectors, and the habitats are marked with geometrical figures: triangle—N1+N2 (natural habitats), circle—A1+A2 (semi-natural habitats). Abbreviations of species names consist of one letter of a generic name and three letters of a species name. All abbreviations as in Figure 4.

**Figure 9 biology-10-01266-f009:**
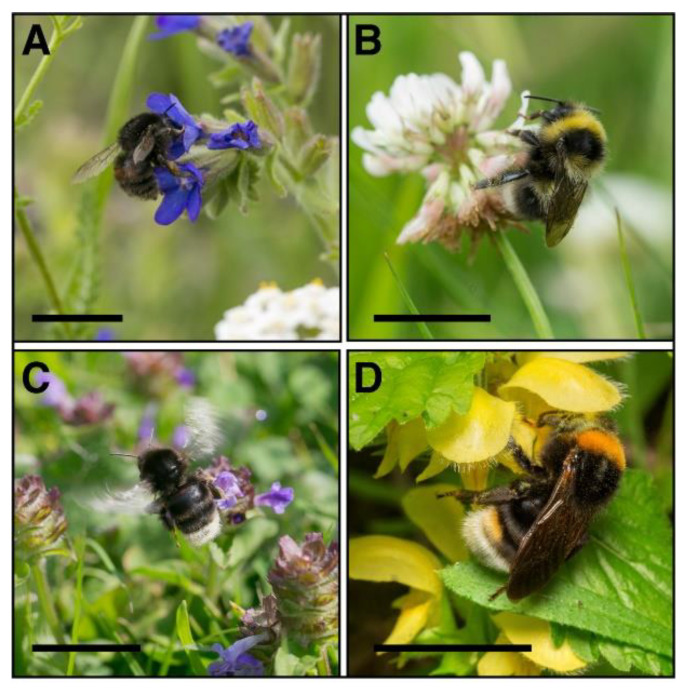
Examples of bumblebee species collected in southern Poland: (**A**) *Bombus humilis*, female visiting *Anchusa officinalis*; (**B**) *Bombus jonellus*, worker visiting *Trifolium* sp.; (**C**) *Bombus soroeensis*, worker visiting *Prunella vulgaris*; (**D**) *Bombus vestalis* from visiting *Galeobdolon luteum.* Scale bars = 0.5 cm.

**Table 1 biology-10-01266-t001:** Summary of the collected bumblebee specimens in the Małopolska Upland: Świętokrzyski National Park (SNP), Cisowsko-Orłowiński Landscape Park (COLP), Chęcińsko-Kielecki Landscape Park (ChKLP), and Nadnidziański Landscape Park (NLP) between the years 2003–2006 and 2017–2020.

SNP			COLP		ChKLP		NLP	
	Species	Specimens	Species	Specimens	Species	Specimens	Species	Specimens
2003	13	76	15	257	15	117	15	93
2004	11	91	19	183	12	150	14	150
2005	15	152	17	516	15	257	16	183
2006	15	179	17	414	17	532	18	321
2017	11	44	14	164	14	95	15	66
2018	10	60	16	150	12	114	18	110
2019	14	97	15	365	15	193	16	106
2020	14	127	14	305	15	321	17	226

**Table 2 biology-10-01266-t002:** The number of bumblebee specimens in natural (n) and semi-natural (sn) habitats in SNP, COLP, NLP, and ChKLP between 2003–2006 and 2017–2020.

		2003/2004	2005/2006	2017/2018	2019/2020
n	COLP	48/59	138/167	123/141	112/118
sn		69/91	119/365	105/225	253/187
n	NLP	23/48	65/166	54/152	36/114
sn		70/102	118/155	102/136	73/112
n	ChKLP	168/123	252/284	240/380	202/214
sn		89/60	264/130	200/125	185/105
n	SNP	16/29	66/81	54/88	41/52
sn		60/62	86/98	76/78	56/75

## Data Availability

The data presented in this study are available on request from the corresponding author.

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
