# Peer review of "Effects of Open and Forest Habitats on Distribution and Diversity of Bumblebees (Bombus) in the Małopolska Upland (Southern Poland): Case Study"

_biology, 2021, doi:10.3390/biology10121266_

Round 1

Reviewer 1 Report

These are my main comments on the manuscript (Biology-1441114) entitled “Effects of Open and Forest Habitats on Distribution and Diversity of Bumble Bees (Bombus) in MaÅ‚opolska Upland (Southern Poland): Case Study”. It is an interesting study examining the diversity and distribution of bumblebees. Such studies provide useful results to know of plant-insect pollinator relationship. Generally, it a well-written study without any serious flaws or mistakes in methodology and presentation. I have suggested some changes in the manuscript. My proposal is to accept it for publication in "Insects" after minor revision.

A few points:

L.10: Sentence starting “Southern Poland…”

L.10: … Bumblebees (Bombus sp.); however,…

L.12: Change “bumble bees” by “bumblebees”. Check in all manuscript.

L.12: Delete Bombus

L.13: …by them. Studies…

L.15: More information about results obtained in this study is needed.

L.19: …influences the abundance…

L.20: Delete “(1) Background:”

L.23: Delete “(2) Methods:”

L.25: Delete “(3) Results:”

L.33: …cover. (4) Conclusions: These studies are necessary…

L.35: Keywords should be in alphabetic order. Also, keywords serve to widen the opportunity to be retrieved from a database. To put words that already are into title and abstracts makes KW not useful. Please choose terms that are neither in the title nor in abstract.

L.48: Bumble bees and bees are…

L.59: Delete “(Apinae: Corbiculata: Bombini)”

L.76: … of worldwide importance…

L.99: Delete “the”

L.101: Place “,” after alnus

L.135: … airtight container with…

L.136: … to prevent the growth of mould? Rephrase

L.142: In each habitat, soil physical (density, structure, and texture), and chemical…

L.144: Change “ml” by “mL”

Ls.151-153: Revise this sentence to eliminate rewordiness.

Ls.163-164: Rephrase this sentence.

L.169: Twenty-five…

L.173: … UTM gird squares…Revise

L.181: … found twenty-one…

L.185: …found twenty-four…

L.186: … the Lamiacae and Fabaceae families.

L.189: … show that the number…

L.190: … sun exposure, and…

L.191-195: Rephrase this sentence

L.196-197: Again, rephrase

L.207: For probability, letter p should be in italic

L.218-219 and 227-228: … The vectors of variables differentiating the…

L:241, 247, and 260: … of one letter of…

L.253 and 265: Change “were” by “was”

L.269: …these groups such as: B. bohemicus,…

L.270: In the semi-natural habitats, a few…

L.271: …B. vestalis) found. The Bombus family prefers open,

L.272: …numerous occurrences of…

L.273-274: Revise this sentence to eliminate rewordiness.

L.302: … species is from…

L.305: … Europe, especially in,…

L.308: Place “,” after Hungary

L.309: Paleartic

L.310: Change “localised” by “localized”

L.313: During our studies, B. soroeensis was found…

L.334: presence or present?

L.339: Change “specialisation” by “specialization”

L.348: Change “are” by “is”

L.362: Change “favour” by “favor”

Ls.366-367: Rephrase this sentence

L.397: …that the climate change poses…

L.403: indicates

Ls.432-435: Rephrase this sentence.

L.451: Twenty-five

L.455: …parks areas are…

Author Response

Answers to points raised by the Reviewers of manuscript ID: biology-1441114

"Effects of Open and Forest Habitats on Distribution and Diversity of Bumblebees (Bombus) in MaÅ‚opolska Upland (Southern Poland): Case Study” by Jolanta BÄ…k-Badowska et al.

I would like to thank you for the remarks about our manuscript titled “Effects of Open and Forest Habitats on Distribution and Diversity of Bumblebees (Bombus) in MaÅ‚opolska Upland (Southern Poland): Case Study”. These comments allowed me to improve the quality of the manuscript. Below, I listed my answers to all questions and points raised by the Reviewers. All changes in revised manuscript are marked in red font.

Answers to the Reviewer’s comments:

Comment

Answer

Reviewer 1

L.10: Sentence starting “Southern Poland…”

Thank you for this comment. I added a correct phrase.

L.10: … Bumblebees (Bombus sp.); however,…

Thank you for this comment. I added a new phrase.

L.12: Change “bumble bees” by “bumblebees”. Check in all manuscript.

Thank you for this comment. I corrected “bumblebees” in all manuscript.

L.12: Delete Bombus

Thank you for this comment. Done.

L.13: …by them. Studies…

Thank you for this comment.

L.15: More information about results obtained in this study is needed.

Thank you for this comment. I added new information in Simple Summary:

“The most frequently observed bumblebee species were: Bombus pascuorum, B. lapidarius, B. pratorum and B. lucorum. The low-numbered bumblebees were: Bombus humilis, B. pomorum, B. veteranus, B. muscorum and B. semenoviellus. There were also four rarely found species: Bombus confusus, B. ruderatus, B. soroeensis and B. jonellus”.

L.19: …influences the abundance…

Thank you for this comment. I deleted “of” and added “the”.

L.20: Delete “(1) Background:”

Thank you for this comment. I deleted “Background”.

L.23: Delete “(2) Methods:”

Thank you for this comment. I deleted “Methods”.

L.25: Delete “(3) Results:”

Done.

L.33: …cover. (4) Conclusions: These studies are necessary…

Thank you for this comment. I deleted “(4) Conclusions” and added “are”.

L.35: Keywords should be in alphabetic order. Also, keywords serve to widen the opportunity to be retrieved from a database. To put words that already are into title and abstracts makes KW not useful. Please choose terms that are neither in the title nor in abstract.

Thank you for this comment and I apologize for the confusion with keywords. I added new keywords:

Bombus; community ecology; conservation; insect pollinators; landscape ecology; Europe”

L.48: Bumble bees and bees are…

Done.

L.59: Delete “(Apinae: Corbiculata: Bombini)”

Thank you for this comment. I deleted “Apinae: Corbiculata: Bombini”.

L.76: … of worldwide importance…

Thank you for this comment. I added a correct word.

L.99: Delete “the”

Done.

L.101: Place “,” after alnus

Done.

L.135: … airtight container with…

Done.

L.136: … to prevent the growth of mould? Rephrase

Thank you for this comment. I deleted this sentence.

L.142: In each habitat, soil physical (density, structure, and texture), and chemical…

Thank you for this comment. I corrected “habitat”.

L.144: Change “ml” by “mL”

Done.

Ls.151-153: Revise this sentence to eliminate rewordiness.

Thank you for this comment. I revised this sentence.

“Changes in sites (in the studied parks), community (between types of communities) and between particular years of the study were tested using statistical significance”.

Ls.163-164: Rephrase this sentence.

Thank you for this comment. I revised this sentence: “Changes in the number of species in samples in the ordination space were presented using diagram”.

L.169: Twenty-five…

Done.

L.173: … UTM gird squares…Revise

Thank you for this comment. I added “transects” instead of “gird squares”. I hope it is  better now.

L.181: … found twenty-one…

Done.

L.185: …found twenty-four…

Done.

L.186: … the Lamiacae and Fabaceae families.

Thank you for this comment. I corrected this phrase.

L.189: … show that the number…

Thank you for this comment. I deleted “in”.

L.190: … sun exposure, and…

Thank you for this comment. I added “comma”.

L.191-195: Rephrase this sentence

Thank you for this comment. I rephrased this sentence:

“We registered a total of 1370 bumblebee specimens in Cisowsko-OrÅ‚owiÅ„ski Landscape Park and 498 specimens in ÅšwiÄ™tokrzyski National Park between 2003-2006. Bumblebees abundance decreased between 2017 and 2020. Overall, we recorded 984 bumblebee specimens in Cisowsko-OrÅ‚owiÅ„ski Landscape Park, and only 328 specimens in ÅšwiÄ™tokrzyski National Park.”

L.196-197: Again, rephrase

Thank you for this comment. I rephrased this sentence:

“For each study sample, the Simpson diversity index varied from 0 (one species in one sample) to 0.895”.

L.207: For probability, letter p should be in italic

Done.

L.218-219 and 227-228: … The vectors of variables differentiating the…

Done.

L:241, 247, and 260: … of one letter of…

Thank you for this comment. I deleted “s”.

L.253 and 265: Change “were” by “was”

Done.

L.269: …these groups such as: B. bohemicus,…

Thank you for this comment. I added “s”.

L.270: In the semi-natural habitats, a few…

Thank you for this comment. I corrected the word “habitats”.

L.271: …B. vestalis) found. The Bombus family prefers open,

Thank you for this comment. I corrected the word “prefers”.

L.272: …numerous occurrences of…

Thank you for this comment. I corrected the word “occurrences”.

L.273-274: Revise this sentence to eliminate rewordiness.

Thank you for this comment. I revised this sentence”

“We observed that the type of habitat influences the overall bumblebee abundance and richness through isolation and changes in food availability, also modifying community composition”.

L.302: … species is from…

Done.

L.305: … Europe, especially in,…

Thank you for this comment. I corrected the word “especially”.

L.308: Place “,” after Hungary

Done.

L.309: Paleartic

Done.

L.310: Change “localised” by “localized”

Done.

L.313: During our studies, B. soroeensis was found…

Thank you for this comment. I added the word “was”.

L.334: presence or present?

Thank you for this comment. It should be “present”.

L.339: Change “specialisation” by “specialization”

Done.

L.348: Change “are” by “is”

Done.

L.362: Change “favour” by “favor”

Done.

Ls.366-367: Rephrase this sentence

Thank you for this comment. I revised this sentence:

“Species such as: B. pascuorum, B. terestris, and B. lucorum occurred in natural and semi-natural habitats, in all studied years.”

L.397: …that the climate change poses…

Thank you for your comment. I deleted “s”.

L.403: indicates

Done.

Ls.432-435: Rephrase this sentence.

Thank you for this comment. I revised this sentence:

“Habitats fragmentation can affect the diversity of bumblebee species and community, and may depend, among other aspects, on their habitat preferences, foraging ranges, and behavioral [46]”.

L.451: Twenty-five

Done.

L.455: …parks areas are…

Done.

Reviewer 2 Report

General comments:

Overall, this paper is a good case study of the bumble bee community in southern Poland. It is a useful contribution to our understanding of bumble bee species occurrence and diversity in Europe, and the authors provide a good discussion of the effects of different types of habitat on bumble bee communities. I have suggested only minor comments below.

Line by line comments:

Line 12: Change “temperature” to “temperate”

Line 13: Delete extra period

Line 19: Delete “of”

Line 21: Add “we” after 2017-2020

Lines 29-30: What does the phrase “Typical and meadow” mean here? Please rewrite this.

Lines 33-34: I suggest minor edits in this sentence, for example: “Studies like this are necessary to anticipate the impact of habitat fragmentation on bumble bees decline.”

Line 45: Change “use” to “require”

Line 51: Should this say “European countries”?

Line 57: Please delete the colon

Line 60: Change “than more” to “more than”

Lines 75-76: I suggest changing these lines to: “…and their conservation is of world-wide importance for preserving plant and animal biodiversity.”

Line 81: I suggest adding to this sentence so it reads: “The scientific monitoring of our current study….” to make it clear that you are talking about the work you did, rather than the work in earlier decades that you reference in the previous sentence.

Line 132: Delete “a combination”

Line 138: Can you add where this university is located? What city?

Lines 146-148: This sentence is talking about the species identification of bumble bee specimens, correct? Can you move it to the end of the previous paragraph?

Line 209: replace “od” with “of”

Lines 279-350: This section of the text should be moved from the Results section to the Discussion, since it is adding extra information and drawing on the literature to provide context for the results.

Line 367: Delete the word “and” at the end of the sentence.

Line 396: Change the word “temperature” to “temperate”

Author Response

Answers to points raised by the Reviewers of manuscript ID: biology-1441114

"Effects of Open and Forest Habitats on Distribution and Diversity of Bumblebees (Bombus) in MaÅ‚opolska Upland (Southern Poland): Case Study” by Jolanta BÄ…k-Badowska et al.

I would like to thank you for the remarks about our manuscript titled “Effects of Open and Forest Habitats on Distribution and Diversity of Bumblebees (Bombus) in MaÅ‚opolska Upland (Southern Poland): Case Study”. These comments allowed me to improve the quality of the manuscript. Below, I listed my answers to all questions and points raised by the Reviewers. All changes in revised manuscript are marked in red font.

Answers to the Reviewer’s comments:

Comment

Answer

Reviewer 2

Line 12: Change “temperature” to “temperate”

Thank you for this comment. I changed it.

Line 13: Delete extra period

Thank you for this comment. I deleted this sentence.

Line 19: Delete “of”

Thank you for this comment. I deleted “of”.

Line 21: Add “we” after 2017-2020

Thank you for this comment. I added “we”.

Lines 29-30: What does the phrase “Typical and meadow” mean here? Please rewrite this.

Thank you for this comment. I corrected the sentence:

 “The meadows from Molinio arrhenatheretea class were characterized by bumblebees communities with a more complex structure”.

Lines 33-34: I suggest minor edits in this sentence, for example: “Studies like this are necessary to anticipate the impact of habitat fragmentation on bumble bees decline.”

Thank you for this comment. I corrected the sentence according to suggestion.

Line 45: Change “use” to “require”

Thank you for this comment. I added “require” according to suggestion.

Line 51: Should this say “European countries”?

Thank you for this comment. I deleted this phrase according to suggestion. I hope that it is much better now.

Line 57: Please delete the colon

Done.

Line 60: Change “than more” to “more than”

Thank you for this comment. I changed it.

Lines 75-76: I suggest changing these lines to: “…and their conservation is of world-wide importance for preserving plant and animal biodiversity.”

Thank you for this comment. I changed this phrase according to suggestion. I hope that it is much better now.

Line 81: I suggest adding to this sentence so it reads: “The scientific monitoring of our current study….” to make it clear that you are talking about the work you did, rather than the work in earlier decades that you reference in the previous sentence.

Thank you for this comment. I added this sentence.

“The scientific monitoring of our current study was aimed at evaluating the distribution and diversity of bumblebees in MaÅ‚opolska Upland (southern Poland) collected during the years 2003-2006 and 2017-2020”.

Line 132: Delete “a combination”

Done.

Line 138: Can you add where this university is located? What city?

Thank you for this comment. I added name of city.

“…(Jan Kochanowski University, Kielce)…”

Lines 146-148: This sentence is talking about the species identification of bumble bee specimens, correct? Can you move it to the end of the previous paragraph?

Thank you for this comment. I moved this phrase according to your suggestion. I hope it is better now.

Line 209: replace “od” with “of”

Done.

Lines 279-350: This section of the text should be moved from the Results section to the Discussion, since it is adding extra information and drawing on the literature to provide context for the results.

Thank you for this comment. I moved this section from the Results section to the Discussion according to suggestion. I revised all references. I hope it is better now.

Line 367: Delete the word “and” at the end of the sentence.

Done.

Line 396: Change the word “temperature” to “temperate”

Done.

With many thanks for all comments.

Sincerely,

Joanna Czerwik-Marcinkowska

Reviewer 3 Report

This study investigated bumble bee occurrence in Southern Poland in two four-year periods over two decades. Studies determining species distributions and densities are important for quantificating the decline in insect pollinators so that solutions can be formulated.

The paper is overly long and can benefit from careful editing for clarity, avoidance of repetition and irrelevance, and better organisation of the material. I could not find statistics supporting some of the statements made in the results and discussion.

Line 41-42: I’m not sure that this sentence accurately reflects their distribution with regards to e.g. Pacific Islands and the distinction between where they are native and have been introduced

Line 42-44: this sentence requires references, although there is overlap with the final sentence of this paragraph.

Line 48-50: Sentence requires greater clarity. Replace ‘related’ with a more appropriate word. Bumble bees does not include bees, it’s the other way around. Unclear if you mean bees or bumble bees are the most important pollinators. Requires referencing.

Line 48-58: I find that this paragraph is largely repeating the previous. I suggest consolidating your topics into different paragraphs rather than covering multiple topics in both paragraphs.

Line 59-61: this information is best placed with your previous information about worldwide distribution

Line 59-72: again this paragraph is covering topics which have been covered in previous paragraphs. I suggest a single topic per paragraph

Line 92: please state the months of the flowering season

Figure 1: would it be possible to show where the four parks appear within the map of Poland?

Line 132: specimens rather than species

Line 179: italics for species

189-191: I am having trouble finding evidence of this and the general trends

Line 279-350 : much of this is better placed in the discussion. Results usually do not refer to other literature, but only the results of the present study.

Line 360-361: I find nothing that quantifies flowering resources

Line 383-384: I don’t find any information relating to climate variables to support this statement

Author Response

Answers to points raised by the Reviewers of manuscript ID: biology-1441114

"Effects of Open and Forest Habitats on Distribution and Diversity of Bumblebees (Bombus) in MaÅ‚opolska Upland (Southern Poland): Case Study” by Jolanta BÄ…k-Badowska et al.

I would like to thank you for the remarks about our manuscript titled “Effects of Open and Forest Habitats on Distribution and Diversity of Bumblebees (Bombus) in MaÅ‚opolska Upland (Southern Poland): Case Study”. These comments allowed me to improve the quality of the manuscript. Below, I listed my answers to all questions and points raised by the Reviewers. All changes in revised manuscript are marked in red font.

Answers to the Reviewer’s comments:

Comment

Answer

Reviewer 3

Line 41-42: I’m not sure that this sentence accurately reflects their distribution with regards to e.g. Pacific Islands and the distinction between where they are native and have been introduced

Thank you for this comment. I deleted this sentence and I added new information. I revised all references. I hope it is better now.

“Although the distribution of bumblebees encompasses a wide geographic range from Arctic tundra to lowland tropical forest, they are clearly most abundant in mountain habitats, the cold and temperate regions of the Northern Hemisphere [1,2].”

Line 42-44: this sentence requires references, although there is overlap with the final sentence of this paragraph.

Thank you for this comment. I added reference.

“As among many wild bees, their abundance and diversity declined in recent years, which may threaten the stability of pollinators communities [11–13].”

[11] Gallai, N.; Salles, J.; Settele, J.; Vaissiere, B.E. Economic valuation of the vulnerability of world agriculture confronted with pollinator decline. Ecological Economics 2009, 68, 810–821.

[12] Leonhardt, S.D.; Gallai, N.; Garibaldi, L.A.; Kuhlmann, M.; Klein, A.M. Economic gain, stability of pollination and bee diversity decrease from southern to northern Europe. Basic and Applied Ecology 2013, 14, 461–471.

[13] Hines, H.; Hendrix, S.D. Bumble bee (Hymenoptera: Apidae) diversity and abundance in tallgrass prairie patches: effects of local and landscape floral resources. Environmental Entomology 2015, 34, 1777–1484.

Line 48-50: Sentence requires greater clarity. Replace ‘related’ with a more appropriate word. Bumble bees does not include bees, it’s the other way around. Unclear if you mean bees or bumble bees are the most important pollinators. Requires referencing.

Thank you for this comment and I apologize for the confusion with bees. I added reference.

“Bumblebees and bees are bonded to flowers by their use of pollen as a protein source, hence they are the most important pollinator species in terrestrial ecosystems [13,14].”

Line 48-58: I find that this paragraph is largely repeating the previous. I suggest consolidating your topics into different paragraphs rather than covering multiple topics in both paragraphs.

Thank you for this comment. I deleted the repeating sentences and I corrected all Introduction section.

Line 59-61: this information is best placed with your previous information about worldwide distribution

Thank you for this comment. I corrected the section according to suggestion. I revised all references. I hope it is better now.

Line 59-72: again this paragraph is covering topics which have been covered in previous paragraphs. I suggest a single topic per paragraph

Thank you for this comment and I apologize for the confusion with line 59-72. I changed this paragraph.

Line 92: please state the months of the flowering season

Thank you for this comment. I added the months of the flowering season.

“The field trips covered the entire flowering season (from the beginning of May to the beginning of September) and we collected bumblebees with an aerial net on flowers or while in flight”.

Figure 1: would it be possible to show where the four parks appear within the map of Poland?

Thank you for this comment. I added the parks in Figure 1 according to suggestion.

Line 132: specimens rather than species

Thank you for this comment. I corrected “specimens”.

Line 179: italics for species

Done.

Line 189-191: I am having trouble finding evidence of this and the general trends

Thank you for this comment. I corrected this sentence.

“The studies show that the number of bumblebees species changed over the years and  availability of flowering resources also changed.”

During the field trip covered the entire flowering season, we collected e.g. 251 plant species from COLP, 279 from NLP, 310 from ChKLP and 243 from SNP (data from one year study). We are preparing another publication with complete data.

Line 279-350 : much of this is better placed in the discussion. Results usually do not refer to other literature, but only the results of the present study.

Thank you for this comment. I moved this section from the Results section to the Discussion according to suggestion. I revised all references. I hope it is better now.

Line 360-361: I find nothing that quantifies flowering resources

Thank you for this comment. I added new information.

“A total of 1083 blooming plants were identified. The species most frequently collected were Picea sp. (46.6% of works), Taraxacum officinale (58%), and Salix sp. (42%).”

Line 383-384: I don’t find any information relating to climate variables to support this statement

Thank you for this comment. We used the meteorological information published in the report (Department of Meteorology and Bioclimatology, Institute of Geography, Jan Kochanowski University in Kielce) and from the synoptic station of the Institute of Meteorology and Water Management in Kielce-Suków.

With many thanks for all comments.

Sincerely,

Joanna Czerwik-Marcinkowska

Reviewer 4 Report

The article “Effects of open forest habitats on distribution and div3ersity of bumble bees in Malopolska upland (Southern Poland): case study” investigates the diversity and distribution of bumble bee species in two natural mixed and riparian forests and two semi-natural habitats in southern Poland. They found that bumble bee communities responded to environmental factors in different ways depending on type of habitat. Forest habitats had the lowest bumble bee abundance, richness, and diversity whereas the highest was seen in open ruderal-segetal habitats. Typical (not sure what is meant by this) and meadow habitats had more complex community structure of bumble bees. Overall, I think the sampling methods, identification and analyses are fine and this data should definitely be shared to the community however, I have major concerns with the overall quality of writing and conclusions that are drawn form the data presented. The manuscript can use significant copy editing for grammar and style. The Manu was filled with highlights which I thought should be more of a rough draft and not something submitted to a journal.  The discussion can use a bit of trimming to keep it short and to the point of the main findings of the paper and their relevance to the current literature in the area. The discussion is a bit long and often draws conclusions that can not be backed up by the data presented. For example, line 361, they claim that during the study, loss of natural flowering resources decreased bumble bee abundance and richness etc. However, there was no quantification of floral resources that I was able to see in the methods or results aside from listing the dominant floral species of the habitats studied. Perhaps this information was left out? If so, it should be added to the paper as it stands these conclusions can not be made with the data presented. With this in mind, I am recommending rejection of this manuscript due to these flaws…I believe it needs more than major revisions, taking more than the normal turnover time for revisions. Please see the attached document for in-text comments.

Author Response

Answers to points raised by the Reviewers of manuscript ID: biology-1441114

"Effects of Open and Forest Habitats on Distribution and Diversity of Bumblebees (Bombus) in MaÅ‚opolska Upland (Southern Poland): Case Study” by Jolanta BÄ…k-Badowska et al.

I would like to thank you for the remarks about our manuscript titled “Effects of Open and Forest Habitats on Distribution and Diversity of Bumblebees (Bombus) in MaÅ‚opolska Upland (Southern Poland): Case Study”. These comments allowed me to improve the quality of the manuscript. Below, I listed my answers to all questions and points raised by the Reviewers. All changes in revised manuscript are marked in red font.

Answers to the Reviewer’s comments:

Comment

Answer

Reviewer 4

Line 50: I am not sure what this sentence is saying

Thank you for this comment and I apologize for the confusion with this sentence. I deleted some repeating sentences and I added new information. I revised all references. I hope it is better now.

Line 62: 40 species have been described

Thank you for this comment. Done.

Line 90: Why a permanent transect? Can you justify why this was done opposed to conducting a different transect each time?

Thank you for this comment. I added new information.

“In each of the sixteen sites, we established a permanent bee walk transect of 100 m to 100 m (squares of the UTM gird) to count foraging bumblebees (a modified version of Goulson et al. [6]).”

Line 100: To reach a broader audience can you include the common names for these plant species if possible?

Thank you for this comment. I added the common names for these plant species.

Quercus robur (common oak), Betula pendula (silver birch), Frangula alnus (alder buckthorn), Corylus avellana (common hazel), Alnus glutinosa (common alder), Filipendula ulmaria (meadowsweet), Equisetum sylvaticum (wood horsetail), Asarum europaeum (asarabacca), Lysimachia vulgaris (yellow loosestrife), Valeriana officinalis (garden valerian), Artemisia vulgaris (common mugwort), Agrostis capillaris (common bent), Cirsium arvense (creeping thistle), Plantago lanceolata (ribwort plantain) and Vicia hirsuta (hairy tare)”.

Line 230: Try to fit this on the same page as the actual table

Thank you for this comment. I corrected Table 1.

Line 233: add this to the other columns in the table

Thank you for this comment. Done.

Line 361: Did you calculate loss of floral resources? I do not recall the quantification of floral resources in this methodology or results.

Thank you for this comment. I added new information in Results section.

“A total of 1083 blooming plants were identified. The species most frequently collected were Picea sp. (46.6% of works), Taraxacum officinale (58%), and Salix sp. (42%).”

During the field trip covered the entire flowering season, we collected e.g. 251 plant species from COLP, 279 from NLP, 310 from ChKLP and 243 from SNP (data from one year study). We are preparing another publication with complete data.

With many thanks for all comments.

Sincerely,

Joanna Czerwik-Marcinkowska

Round 2

Reviewer 3 Report

I appreciate the work the authors have put into improving the manuscript. However I still have major concerns about the conclusions made given a lack of data. You cannot make statements without providing proof. If that analysis is part of another paper then these claims should be made in that paper and not in this one. Every result must be justified and all methods given. While more information has been provided, it is not sufficient to justify the statements made e.g. a breakdown of plants per year, statistical tests involving climate etc.

Comment on how your sampling may have led to a reduction in population

I find the discussion overly long and repetitive.

Line 19: italics for B. jonellus

Line 50: reference

Line 51: replace ‘their’ with ‘bumblebee’

Line 53: remove ‘from’

Line 58-59: suggest making this a different sentence as it about central Europe while the start and end of the sentence are worldwide

Line 63: ‘countries which lacks comprehensive data on’

Line 65: 1960s and 1980s

Figure 1d: site 2 seems to be outside the park?

Line 111: and 2017-2020?

Line 122: remove ‘During the collecting’

Line 133: properties were studied

Line 136: what was analysed?

Line 139: what changes in sites? It makes it sound as if you changed which sites you used but I think that is not the case and you are more likely to mean changes in Simpson diversity index. You test for statistical significance, you don’t test using statistical significance. Suggest instead ‘One-way ANOVA was used to test for changes in Simpson diversity at sites across years’.

Line 143: replace ‘done’ with ‘calculated’

Line 147: explain what “0-1” system is

Line 149: replace ‘done’ with ‘performed’

Line 168: we found

Line 169 move ‘of bumblebees’ after ‘species’

Line 170: we observed twelve bumblebee species in 2004

Line 172: remove bumblebees.

Line 173: plants

Line 174: replace ‘have chosen’ with ‘were more commonly found on the’

Line 174: Elsewhere you have written out numbers

Line 177: unclear what you mean by ‘works’. Are these just the plants that were flowering, or are these the plants the specimens were collected off?

Line 177-179: can you qualify this statement with evidence? Without evidence you cannot make this claim.

Line 185-186: move reference to the figure to the previous sentence and remove this sentence

Line 187: add ‘for either year, community or study site’

Line 187: remove ‘both’

Line 189: replace ‘insect’ with ‘bumblebees’. Do you mean in terms of number or diversity?

Line 199: number of species; ‘is significantly lower.’

Line 251: place B. before norvegicus and pratorum; remove ‘were’

Line 259: were found

Line 262: I haven’t seen anything detailing food availability, or plant community composition

Figure 9: I think unneccesary

Line 271: 255 bumblebee specimens. This sentence needs revision.

Line 281: I find no data on loss of natural flowering resources

Line 288: I find no data on flower density

Line 289: who are ‘they’

Line 306: remove ‘were’; sentence needs work

Line 312: remove ‘were’

Line 316: ‘were found to be’

Line 323: the species is endangered rather than the distribution

Line 339: is a small species

Line 341: more particular what?

Line 344-346: there is no data giving how many specimens were collected on which species of plants

Line 349: ‘tend’ not ‘trend’

Line 375: We found

Line 375: There is no analysis of climatic variables presented

Line 389: that climate

Line 394: This study

Line 430: sentence needs work

Table S1: explain period

Author Response

Answers to points raised by the Reviewer of manuscript ID: biology-1441114

"Effects of Open and Forest Habitats on Distribution and Diversity of Bumblebees (Bombus) in MaÅ‚opolska Upland (Southern Poland): Case Study” by Jolanta BÄ…k-Badowska et al.

I would like to thank you for the remarks about our manuscript titled “Effects of Open and Forest Habitats on Distribution and Diversity of Bumblebees (Bombus) in MaÅ‚opolska Upland (Southern Poland): Case Study”. These comments allowed me to improve the quality of our manuscript. Below, I listed my answers to all questions and points raised by the Reviewers. All changes in revised manuscript are marked in green font.

Answers to the Reviewer’s comments:

Comment

Answer

Reviewer 3

L.19: italics for B. jonellus

Thank you for this comment. I changed it.

Line 50: reference

Thank you for this comment. I added a new reference [4].

Line 51: replace ‘their’ with ‘bumblebee’

Thank you for this comment. I replaced “their” with “bumblebee”.

Line 53: remove ‘from’

Thank you for this comment. Done.

Line 58-59: suggest making this a different sentence as it about central Europe while the start and end of the sentence are worldwide

Thank you for this comment. I added a correct phrase.

„The numbers of bumblebee species are declining in Europe, North America, and Asia due to a number of factors, including changes in the land-use (reduced food plants), loss of nesting habitats and climate change effects [15-17]. This effect was also observed in boron mixed (Vaccinio-Piceetea) and riparian forest (Querco-Fagetea) which are some of the most valuable and endangered ecosystems in Central Europe, and their conservation is of worldwide importance for preserving plants and animals biodiversity [18].”

Line 63: ‘countries which lacks comprehensive data on’

Thank you for this comment. I added a correct phrase.

Line 65: 1960s and 1980s

Thank you for this comment. I added a correct phrase.

Figure 1d: site 2 seems to be outside the park?

Thank you for this comment. I corrected Figure 1d.

Line 111: and 2017-2020?

Thank you for this comment. Done.

Line 122: remove ‘During the collecting’

Thank you for this comment. I deleted this phrase.

Line 133: properties were studied

Thank you for this comment. I added “were”.

Line 136: what was analysed?

Thank you for this comment. I deleted this phrase.

Line 139: what changes in sites? It makes it sound as if you changed which sites you used but I think that is not the case and you are more likely to mean changes in Simpson diversity index. You test for statistical significance, you don’t test using statistical significance. Suggest instead ‘One-way ANOVA was used to test for changes in Simpson diversity at sites across years’.

Thank you for this comment and I apologize for the confusion with line 139. I added new sentence according to suggestion.

Line 143: replace ‘done’ with ‘calculated’

Thank you for this comment. I replaced the words.

Line 147: explain what “0-1” system is

Thank you for this comment. I added new information.

“The writing “0-1” in the primary data table means sample coming from a site/year/community and is marked as “1” in the column indicating the site/year/community. While in the other columns of this table the same sample will be marked as “0” because it does not belong to the indicated variable”.

Line 149: replace ‘done’ with ‘performed’

Thank you for this comment. I replaced the words.

Line 168: we found

Done.

Line 169 move ‘of bumblebees’ after ‘species’

Thank you for this comment. I corrected this phrase.

Line 170: we observed twelve bumblebee species in 2004

Thank you for this comment. I corrected this sentence.

Line 172: remove bumblebees.

Done.

Line 173: plants

Done.

Line 174: replace ‘have chosen’ with ‘were more commonly found on the’

Thank you for this comment. I replaced this sentence.

Line 174: Elsewhere you have written out numbers

Thank you for this comment. I added number “sixty-five”.

Line 177: unclear what you mean by ‘works’. Are these just the plants that were flowering, or are these the plants the specimens were collected off?

Thank you for this comment. I added new information:

“…(46.6% of plants the specimens were collected),….”

Line 177-179: can you qualify this statement with evidence? Without evidence you cannot make this claim.

Thank you for this comment. I added new information.

“The studies show that the number of bumblebees species changed over the years and based on phytosociological and ecological analysis of plant communities the flower richness also changed (for plant species we estimated flower richness as the average number of flowering species per sampling day and site).”

Line 185-186: move reference to the figure to the previous sentence and remove this sentence

Thank you for this comment. I corrected this sentence.

Line 187: add ‘for either year, community or study site’

Done.

Line 187: remove ‘both’

Done.

Line 189: replace ‘insect’ with ‘bumblebees’. Do you mean in terms of number or diversity?

Thank you for this comment. I replaced this words. “….the number of bumblebees”

Line 199: number of species; ‘is significantly lower.’

Thank you for this comment. Done.

Line 251: place B. before norvegicus and pratorum; remove ‘were’

Thank you for this comment. Done.

Line 259: were found

Done.

Line 262: I haven’t seen anything detailing food availability, or plant community composition

Thank you for this comment. I deleted this sentence.

Figure 9: I think unneccesary

Figure 9 shows interesting bumblebee species and flowering plants collected in southern Poland.

Line 271: 255 bumblebee specimens. This sentence needs revision.

Thank you for this comment. I corrected this sentence.

“Two hundred and fifty-five specimens of bumblebee were found in natural habitats in 2003, while in 2004, two hundred and fifty-nine specimens and five hundred and twenty-one specimens of bumblebee in 2005, whereas in 2006, six hundred and ninety-eight specimens (Table 2). While in the year 2017, three hundred and sixty-nine specimens of bumblebee were identified and in the following years of the study, a decline in the bumblebee communities were observed.”

Line 281: I find no data on loss of natural flowering resources

Thank you for this comment. I corrected this sentence.

“During this eight year study we observed that the number of flowering plants and bumblebees abundance and richness decreased as composition of communities changed, by reducing habitat specialized species in favour of highly generalist one.”

Line 288: I find no data on flower density

Thank you for this comment. I corrected this sentence.

“For bumblebees species, in study habitats, the number of flowering plants was important, as it is positively related to their abundance and richness.”

Line 289: who are ‘they’

Thank you for this comment. Bumblebees.

Line 306: remove ‘were’; sentence needs work

Thank you for this comment. I corrected this sentence.

“The community of bumblebees occurring in MaÅ‚opolska Upland (southern Poland) needs permanent scientific monitoring”.

Line 312: remove ‘were’

Done.

Line 316: ‘were found to be’

Done.

Line 323: the species is endangered rather than the distribution

Thank you for this comment. I corrected this sentence.

Line 339: is a small species

Done.

Line 341: more particular what?

Thank you for this comment. I corrected this sentence.

Line 344-346: there is no data giving how many specimens were collected on which species of plants

Thank you for this comment. I corrected this sentence.

Line 349: ‘tend’ not ‘trend’

Done.

Line 375: We found

Done.

Line 375: There is no analysis of climatic variables presented

Thank you for this comment. I deleted this sentence.

Line 389: that climate

Done.

Line 394: This study

Done.

Line 430: sentence needs work

Thank you for this comment. I corrected this sentence.

I find the discussion overly long and repetitive.

Thank you for this comment. I deleted repetition and I corrected section Discussion according to suggestion. I hope that it is much better now.

Table S1: explain period

Thank you for this comment. I corrected for “months”.

With many thanks for all comments.

Sincerely,

Joanna Czerwik-Marcinkowska

Reviewer 4 Report

The authors have made significant changes to the text improving the quality of the manuscript. It appears that the other reviewer's comments have been responded to as well.   

Author Response

I would like to thank you for the remarks about our manuscript titled “Effects of Open and Forest Habitats on Distribution and Diversity of Bumblebees (Bombus) in MaÅ‚opolska Upland (Southern Poland): Case Study”. These comments allowed me to improve the quality of our manuscript. 

With many thanks for all comments.

Sincerely,

Joanna Czerwik-Marcinkowska
